# StructFormer: Document Structure-based Masked Attention and its impact on Language Model Pre-Training

## Abstract

Most state-of-the-art techniques for Language Models (LMs) today rely on transformer-based architectures and their ubiquitous attention mechanism. However, the exponential growth in computational requirements with longer input sequences confines Transformers to handling short passages. Recent efforts have aimed to address this limitation by introducing selective attention mechanisms, notably local and global attention. While sparse attention mechanisms, akin to full attention in being Turing-complete, have been theoretically established, their practical impact on pre-training remains unexplored. This study focuses on empirically assessing the influence of global attention on BERT pre-training.

The primary steps involve creating an extensive corpus of structure-aware text through arXiv data, alongside a text-only counterpart. We carry out pre-training on these two datasets, investigate shifts in attention patterns, and assess their implications for downstream tasks. Our analysis underscores the significance of incorporating document structure into LM models, demonstrating their capacity to excel in more abstract tasks, such as document understanding.

## 1 Introduction

Given a universal set of Vocabulary $\mathcal{V}$, the primary objective of a Language Model (LM) is to learn a distribution for a sequence of words, denoted as $P(w_1, w_2, w_3, \ldots, w_n)$, where each $w_i$ belongs to the set $\mathcal{V}$. By building a model based on this word distribution, we can calculate the probability of the next word occurring in a given sequence, expressed as $P(w_n|w_1, w_3, \ldots, w_{n-1})$. Typically, the word with the highest probability is selected as the next word, i.e., $\arg\max_n P(w_n|w_1, w_3, \ldots, w_{n-1})$. Until recently, memory-aware deep learning methods like LSTM, BiLSTM, and other sequential models were the go-to choices for learning the underlying distribution of training data.

However, the landscape of Language Models has evolved with the introduction of the Transformer-based architecture, as initially proposed in Vaswani et al. (2017). Recent state-of-the-art (SOTA) techniques now rely exclusively on Transformers, leveraging their ubiquitous attention mechanism. In the case of attention-based Transformer models, each word learns a self-attention score concerning every other word in the vocabulary $\mathcal{V}$, effectively capturing the relationships between words in a given corpus.

While these Transformer models have been highly successful, a fundamental challenge lies in their computational and memory demands. The attention mechanism scales quadratically in terms of both memory and computation. This limitation makes applying attention to an entire document both expensive and challenging, effectively restricting the application of Transformers to handling only short passages. To mitigate this limitation, researchers have proposed a sparse-attention mechanism Beltagy et al. (2020a). Various methods have emerged to implement this sparse-attention mechanism Ainslie et al. (2020), Zaheer et al. (2020b), with a common approach being the division of attention into two categories: local and global. In the case of local attention, tokens attend to their nearby neighbors within a defined distance $k$, whereas global tokens focus on a selective subset of tokens ($l \ll k$) and are subsequently attended to by all other tokens within an input sequence. This division effectively curbs the computational cost, leading to a linear increase in proportion to the combined size of local and global attention.

Local attention is conceptualized as a sliding window, where a token at position $n$ attends to its surrounding tokens within a window of size $w$. This concept is rooted in human reading behavior, where attention predominantly centers on the paragraph or section being actively read. To encompass the entire context, documents are traditionally structured into chapters, sections, and subsections, each identified by titles and subtitles.

Despite the documented superiority of sparse-attention-based models over their dense attention counterparts Beltagy et al. (2020b), Ainslie et al. (2020), Zaheer et al. (2020b), an area that has yet to be thoroughly explored is the impact of global tokens during the pre-training phase. Existing models are typically trained exclusively with local tokens within a window Beltagy et al. (2020a), and global tokens are brought into play only during downstream tasks.

The present work seeks to address this gap in knowledge by conducting an in-depth examination of the impact of global tokens on the pre-training of Language Models (LMs). A significant challenge in integrating global tokens into pre-training lies in preparing a sufficiently large pre-training corpus of text with properly identified global tokens. Drawing inspiration from PubLayNet Zhong et al. (2019) and DocBank Li et al. (2020), we overcome this challenge by generating structure-aware documents from LaTeX files sourced from arXiv Ginsparg (2011). Within this corpus, we employ LaTeX document structures to identify titles, headings, and sub-headings, which are then incorporated as global tokens during the pre-training process.

Moreover, the significance of incorporating document structure into the pretraining phase of Language Models cannot be overstated. While a vast corpus of text data is essential for language models to learn the intricacies of language, it often lacks the structured semantic information that is crucial for understanding the context and relationships within a document. Document structure, including titles, headings, sub-headings, and hierarchical organization, inherently encodes a wealth of semantic meaning. Titles provide a concise summary of the document's main topics, while headings and sub-headings guide readers through the document's content, offering a roadmap for understanding the document's context and logical flow. By introducing this structured information into the pretraining process, LMs have the potential to grasp not only the nuances of language but also the underlying organizational and semantic structure within documents, enhancing their capacity to tackle complex tasks such as document understanding and abstract summarization. The integration of structural awareness aligns with the broader goal of enabling LMs to bridge the gap between raw text data and meaningful, context-aware language processing.

The subsequent sections of this paper provide a detailed account of this approach and present the outcomes of this comprehensive examination. This research seeks to illuminate the impact and potential benefits of integrating global tokens into the pre-training phase of LMs, thus contributing to the broader understanding of efficient and effective language modeling approaches. Further, the method acts as a proxy for incorporating document structure in pre-training. The change in pre-training influences the attention pattern in a positive way, by capturing stronger relations between keywords and section headers. Our experiments show that employing this method for BERT helps significantly in downstream tasks and provides evidence that learning during pre-training can go beyond natural language understanding (NLU).

## 2 Related works

Kevin Clark et al. did an analysis of BERT's attention Clark et al. (2019) focusing on the analysis of the attention mechanisms of pre-trained models. Though our work provides the method, data, and pre-trained models for the analysis, examining the outputs of language models on carefully chosen input sentences is out of the scope of our current work. We keep the suggested analysis for future work. Jesse Vig provides open-source tools Vig (2019) that can be used to visualize attention at multiple scales, each of which provides a unique perspective on the attention mechanism. They have demonstrated the tool on BERT and GPT-2 models and present three example use cases: detecting model bias, locating relevant attention heads, and linking neurons to model behavior. Another line of work Adi et al. (2016); Giulianelli et al. (2018); Zhang & Bowman (2018) investigates the internal vector representations of the model often using probing classifiers. Again the line of work is on their linguistic abilities of models without explicitly being trained for the tasks. We keep it for our future work.

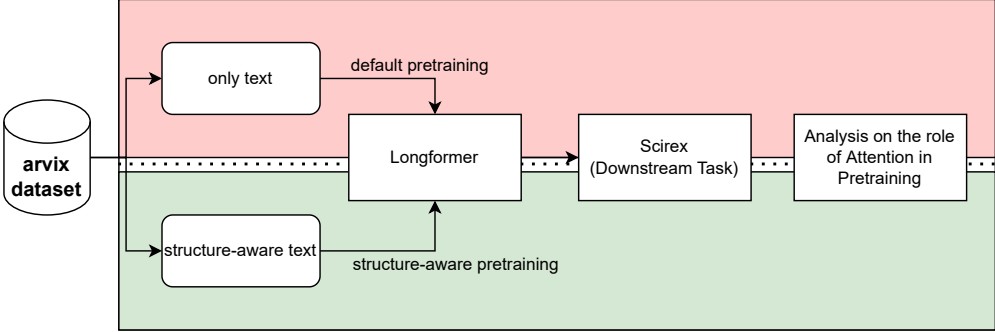

Figure 1: Illustration of our approach to empirically analyzing masked attention during the pre-training process

The following works use document structure for various tasks:

1. **Longformer**: Longformer Beltagy et al. (2020b) is another transformer-based model that introduces a new attention mechanism that scales linearly with sequence length, making it easier to process long documents of thousands of tokens or longer. The attention mechanism combines local windowed attention with task-motivated global attention, allowing for the building of contextual representations of the entire context using multiple layers of attention. During the pretraining of Longformer, the non-availability of global attention pertaining can hinder its ability to capture broader contextual relationships across distant parts of a text. This sparse use of global attention might lead to potential blind spots, where the model may miss essential contextual cues that could be pivotal for certain tasks.

2. **HEGEL**: HEGEL Zhang et al. (2022) uses Hypergraph Transformer which can take longer context and utilizes it for long document summarization. HEGEL addresses the intricacies of high-order cross-sentence relations, offering a novel approach to update and refine sentence representations through the application of hypergraph transformer layers. However, it is exclusive to text summarization and does not help models understand inherent document structure for other tasks.

3. **HIBRIDS**: HIBRIDS Cao & Wang (2022) uses hierarchical biases to encode document structure and then uses this to compute better attention scores. The work introduces a new task: hierarchical question-summary generation, aimed at summarizing salient content within source documents into a hierarchy of questions and summaries, with each follow-up question seeking to delve into the content of its parent question-summary pair. The model is able to outperform similar models in the quality of hierarchical structures and the extent of content coverage.

The works above provide compelling evidence that document structure and semantic organization are valuable in natural language tasks. However, these approaches primarily employ document structure during the fine-tuning process without the ability to adapt to it. Instead, they utilize it as a fixed component. Our approach aims to bridge this gap by introducing a novel method for pre-training BERT-based models. This method enables the models to seamlessly integrate document structure into their understanding of natural language, allowing them to learn and adapt to document structures for more contextually aware language processing.

## 3 Methodology

Our work tries to develop a way to learn the semantic information of document structure in pre-training by using sparse attention. For this purpose, our goal is to analyze the impact of global tokens on pre-training. The methodology for doing this can be divided into the following parts

1. **Pre-training corpus**: We need a large corpus of text data for pre-training that is representative of our target task. We should be able to use the same corpus with and without global tokens. We used the large corpus of latex files available from `https://arxiv.org/`. We prepare two parallel corpora, one that is the default which has only text, and another corpus that is structure-aware. These two data are used for the two pertaining experiments. More details about data preparation are in Section 4

2. **Model architecture**: From a list of sparse attention models Child et al. (2019); Beltagy et al. (2020a); Zaheer et al. (2020a), for the purpose of our work, we chose Longformer, a modified Transformer with an attention mechanism as a combination of a windowed local-context self-attention and an end task-motivated global attention that encodes inductive bias about the task Beltagy et al. (2020a). Longformer has been shown to be effective against many traditional dense attention-based transformers on many NLP tasks with better pre-training metrics yet simple to modify and use for our purpose. The model is a modification of BERT, and is used for downstream tasks where there is an inherent need for global attention for some tokens. STRUCTFORMER model uses Longformer architecture and utilizes global attention for header tokens in the pre-training process.

3. **Pre-training**: We use masked language modeling (MLM) as our pre-training task. The model tries to predict masked token values based on the neighboring tokens. However, here our contribution is to use the local window as is and special global tokens for the headers which now also attend to the masked and the neighboring tokens in the window. This results in the model understanding language as well as the document structure. It is interesting to note that this method can be extended to any BERT-based model since any such model can be converted to a sparse attention transformer.

4. **Attention Patterns**: To test our claim that global tokens during pre-training help language models identify document structure, we analyze the attentions of STRUCTFORMER model. We then compare this against the longformer model trained without global tokens. We observe that STRUCTFORMER model shows significantly higher attention scores between keywords and header tokens as against vanilla longformer. The attention patterns confirm that the model learns not just natural language, but also the structure of documents by identifying the header tokens. A special dataset is required for this result, for which we need keywords around header tokens of documents. This dataset is another contribution to our work.

5. **Generalizability**: Since our pre-training dataset comprises of scientific documents only, a natural question arises about the ability of the model to generalize for other downstream tasks. While the goal of STRUCTFORMER model is not to replace BERT as a foundation model, we demonstrate that it is able to generalize well for various downstream tasks. Moreover, even with structure-aware pre-training, the model does not show decrease in performances for non-document downstream tasks, against vanilla pre-training. We use the Wang et al. (2018) dataset to compare the generalizability of our method.

6. **Downstream task**: We use our model for SciREX (Scientific REpresentation eXtraction) downstream task Jain et al. (2020) which is a document-level IE dataset that encompasses multiple IE tasks, including salient entity identification and document-level N-ary relation identification from scientific articles. The SciREX benchmark dataset can be used to evaluate models for scientific representation learning, majorly focusing on the automatic extraction of structured representations from scientific documents. The problem can be broken down by first identifying named entities, then clustering the named entities and recognizing the salient mentions in the clusters. Finally, the relation tuples are extracted, and SciREX attempts to do all this using a single model. Since SciREX data is also derived from the arxiv, studying the effect of global tokens becomes a lot easier and more relevant. The only modification needed is to change the pre-trained BERT with STRUCTFORMER pre-trained model to finetune on SciREX.

The entire workflow of our controlled experiment is visualized in Figure 1.

Table 1: Statistics on extracted document

|  | Tokens | Headers | Tokens per header |
|---|---|---|---|
| Minimum | 2 | 1 | 1 |
| Maximum | 4,553,287 | 498 | 40,592 |
| Mean | 15,266 | 14 | 106 |
| SD | 31,993 | 9 | 204 |

## 4 Dataset Generation

The necessity of pre-training arises from the significant shift in context when introducing structure into language modeling. Currently, no pre-existing datasets suitable for language model (LM) pre-training can adequately represent the structure of input data. We took inspiration from DocBank Li et al. (2020), opting to utilize LaTeX documents from arxiv Ginsparg (2011), as the structure of these documents can be easily extracted from their LaTeX codes.

For the purpose of structure extraction, we downloaded a total of $1,129,787$ LaTeX documents from `https://arxiv.org/`, covering the period from 2000 to 2018. We combined all these documents into a single flattened file, after eliminating all comments, figures, tables, and equations. The LaTeX codes were processed individually, first extracting the title and abstract, followed by sections, subsections, subsubsections, and paragraphs. While performing this extraction, we ensured the removal of all LaTeX syntax except for indications of bold, italic, and underlined text. TexSoup, a Python package designed for searching, navigating, and modifying LaTeX code, was employed for the extraction process.

During the extraction process, each word was classified as a title word or a paragraph word. Once these words were converted into tokens, their classification was preserved. This was achieved by creating a tuple of the token and a special character denoting whether it was a title token. Special Pytorch data loaders Zolnouri et al. (2020) were created to parse these documents and provide each token and optional token masks for the global token experiment.

The extracted structure was stored in a text file in a specific format. This format comprised a sequence of numerical meta-information followed by the actual words of the node. Each word was accompanied by three Boolean values, representing whether the word was bold, italic, or underlined. Figure 4 and 5 illustrate the extraction process and tokenization. These text files were subsequently used for creating pickle files that form the dataset.

For every ten documents, a single pickle file was generated, containing an iterative list of dictionaries, with each dictionary representing a document. These dictionaries contained three keys: 'title', 'content', and 'sub-levels'. At the first level, the 'title' key stores the document's title, the content' key stores the abstract, and the 'sub-levels key stores a list of dictionaries representing sections, subsections, subsubsections, and corresponding paragraphs. This pattern was maintained till the last level. Thus, we were able to generate a structure-aware corpus suitable for pre-training the model.

Various statistical analyses are performed on the above-created text corpus. For example, min, max, mean and standard deviations were calculated for the number of tokens, number of headers, and number of tokens per header to help decide the sequence length. This is depicted in Table 1.

## 5 Experiments

### 5.1 Structure-aware Pre-training

The Longformer model chosen for training was **Allen AI's 4096_base** Beltagy et al. (2020a). The documents were filtered such that each document contained between $2,000$ to $12,000$ tokens to avoid outliers. Finally, the model was pre-trained on $100,000$ documents. Both the baseline and global token pre-training

were run using this corpus. When storing the documents, the header is recursively identified and its content is stored in the following tokens. This helps identify the various headers in the document inducing a sense of the structure of the document. This is exploited in the pre-training as the tokens corresponding to these headers are used as global tokens, by setting their mask to 1, while the other tokens have a mask set to 0. The local attention window size is set to 256 tokens. Another model with the same architecture was trained with the global attention mask set to 0 for all tokens. This will give us two similar models where the only difference is the structure-aware pre-training. We chose Masked Language Modelling (MLM) Wettig et al. (2022) for the pre-training task. Both the models were pre-trained on the entire corpus of data for the MLM task for 9,000 runs. The local attention window was set to 256. The models took 16 hours each for pre-training. The results obtained on the *bits-per-character (BPC)* metric for the two models is presented in Table 2. The default pre- training's BPC of **2.3** shows the difficulty of the model in understanding the arxiv corpus. As arxiv documents are written in latex documents, in the final output many NLP grammar rules will break such as abrupt sentence shifts between the titles and the first line in the paragraph succeeding it, whereas given the structural information slightly reduces the BPC. This could be because we were able to instruct the model on the difference between titles and paragraphs which can help learn attention better within the sliding window.

## 5.2 Attention Patterns and Visualization

To further explore the impacts of structure-aware pretraining, we examined the attention patterns between our proposed model (referred to as STRUCTFORMER) and a standard vanilla model trained without global tokens.

To conduct this study, we curated a novel dataset from scientific documents published on arxiv in 2023. These documents were never seen by the models during training. The dataset consists of section headings and a few subsequent sentences from each section. We then manually annotated keywords in these sentences which we deemed critical for preserving the context throughout the document.

Our evaluation involved examining the attention between the section headings and these key annotated words in the sentences. We computed the average attention score for these critical relationships across both models. This investigation allowed us to observe and compare the influence of structural pretraining on how the models distribute attention across the document. The results indicate that STRUCTFORMER model shows an increase of more than **20 %** between key-words and header tokens.

The attention patterns in the vanilla model were relatively uniform, with the model apportioning equal attention to all words within the context window. However, there was a notable bias towards the most recent words, showcasing a typical recency effect in language processing tasks.

Conversely, the attention patterns of the STRUCTFORMER model were markedly different. The model demonstrated a variable distribution of attention, concentrating more on specific tokens pertinent to the prediction of the next word. Importantly, the global tokens (representing titles and headings) consistently received high attention scores, signifying that the model effectively harnessed the structure-aware pre-training. This suggests that the STRUCTFORMER model could create a more informed understanding of the document's overall context, a vital aspect in various language understanding tasks. Figure 3 shows an example of an attention pattern for the last transformer layer. "Introduction" is the header token and the rest of the tokens are keywords in the local window. As can be seen, the header token attends more to keywords in structure-aware pre-training.

In essence, our exploration of attention patterns corroborates the potential benefits of structure-aware pre-training. Such a method alters the model's perception and processing of text, encouraging it to focus on global contextual cues and understand the intrinsic structure of the document. As a result, the model could exhibit improved performance across a range of language understanding tasks.

## 5.3 SciREX Fine-tuning

The SciREX dataset was chosen to analyze the effects of structure-aware pre-training in documents. SciREX Jain et al. (2020) (Scientific REpresentation eXtraction) is a benchmark dataset used to evalu-

ate models for scientific representation learning. Since the models were pre-trained on arxiv which is also a scientific document dataset, the contextual information would be better utilized in a SciREX-like dataset.

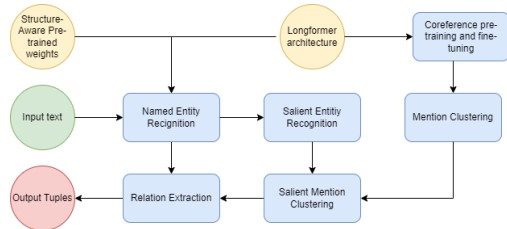

Figure 2: SciREX pipeline with structure-aware corpus pre-trained longformer

Fine-tuning was done by modifying the SciREX training pipeline. In place of BERT, a Longformer model was initialized and the pre-trained weights were loaded. In this analysis, we will consider three different types of models

- **StructFormer**: This is our proposed longformer model which exploits the global token to have structure awareness while pretraining on a large corpus of scientific documents

- **Vanilla Longformer**: This LongFormer model is pre-trained on the same corpus of scientific documents and is a part of our ablation study

- **SciREX Baseline**: This is the model proposed by the authors of SciREX dataset to serve as a baseline

Finally, we present our results on the SciREX dataset task of our model against the other two models. Firstly, in Table 3, we compare the StructFormer model against the SciREX baseline for the end-to-end predicted input.

| Masked Language Modelling Results | |
|---|---|
| Model | Test |
| Structure-Aware pre-training | 2.2136 |
| Default pre-training | 2.3051 |

Table 2: BPC on held-out arxiv test for the different models

| End-to-end (predicted input) | | | | | | |
|---|---|---|---|---|---|---|
| | StructFormer | | | SciREX Baseline | | |
| Task | Precision | Recall | F1 | Precision | Recall | F1 |
| Salient Clusters | **0.2581** | 0.61271 | **0.3419** | 0.2230 | 0.6000 | 0.3070 |
| Binary Relations | 0.0550 | **0.5100** | 0.0890 | **0.0650** | 0.4110 | 0.0960 |
| 4-ary Relations | 0.0019 | **0.2760** | 0.0037 | **0.0070** | 0.1730 | **0.0080** |

Table 3: Summary of the results of StructFormer on the SciREX end-to-end task against the baseline

As can be seen from the results in Table 3, STRUCTFORMER demonstrates a distinct improvement in salient clustering compared to the SciREX baseline. However, this improvement is not reflected in good n-ary relation extraction scores. As explained by the authors in the paper Jain et al. (2020), this is primarily due to the identification of numerous outlier clusters by the end-to-end model, leading to poor subsequent performance. Nevertheless, StructFormer outperforms in salient mention clustering and all tasks leading up to it. It is important to note that this improvement cannot be solely attributed to contextual pre-training. One reason for this is that the SciREX pipeline utilizes a co-reference model, which is pre-trained and fine-tuned on a scientific document corpus. Hence, the improvement can be attributed to either structure-aware pre-training or the Longformer architecture.

| End-to-end (predicted input) | | | | | | |
|---|---|---|---|---|---|---|
| | StructFormer | | | Vanilla Longformer | | |
| Task | Precision | Recall | F1 | Precision | Recall | F1 |
| Salient Clusters | **0.2581** | **0.6127** | **0.3419** | 0.2371 | 0.5949 | 0.3182 |
| Binary Relations | **0.0550** | **0.5100** | **0.0890** | 0.0470 | 0.4906 | 0.0740 |
| 4-ary Relations | **0.0019** | **0.276** | | 0.0013 | 0.2745 | 0.0025 |
| | | | **0.0037** | | | |

Table 4: Comparative results between StructFormer Vanilla Longformer models on Scirex (predicted)

To discern which of the two factors has a greater impact, we conducted an ablation study by pre-training the Longformer model without global tokens, thereby losing the context of structure. The results of this study are summarized in Table 4.

The results from Table 4 clearly demonstrate that structure-aware pre-training significantly contributes to the identification of salient clusters and all the preceding steps in the pipeline. This is evident by the superior performance of the structure-aware pre-training approach across all metrics when compared to the regular model on the predicted input test set. Interestingly, the results of Vanilla Longformer are not considerably different from the SciREX baseline for salient clustering. This ablation study highlights the role of structure-aware pre-training in enabling the model to learn attention patterns that enhance its performance on the predicted input task, surpassing the baseline performance in the SciREX task.

## 5.4 GLUE Fine-tuning

The General Language Understanding Evaluation (GLUE) dataset is an extensive benchmark made up of different tasks related to natural language comprehension. These tasks cover a broad spectrum of language problems, including as sentiment analysis, textual similarity evaluation, paraphrase detection, and judgments of grammaticality.GLUE offers a consistent approach for assessing how well models perform on various language comprehension tasks.

We evaluate our pre-trained models along with BERT on CoLA, MRPC, RTE, STSB, QNLI and SST2 tasks. The comparison between BERT helps us understand about the effect of pre-training on document corpus, while the ablation study between STRUCTFORMER and Vanilla pre-trained Longformer helps us establish the effect of structure-aware pre-training and the effect of global tokens.

The experiments were carried out by setting a batch size of 16 with a learning rate of $2e^{-5}$, for 3 epochs (standard hyper-parameters used for BERT fine-tuning). The results of the fine-tuning are summarized in Table 5. The results indicate that there is not much difference between the three models in all the tasks

| Task | Metric | Vanilla Longformer | StructFormer | BERT base |
|---|---|---|---|---|
| CoLA | Mathews Correlation | 0.502 | 0.469 | **0.521** |
| STSB | Combined Score | **0.871** | 0.856 | 0.858 |
| MRPC | Accuracy | 0.870 | **0.900** | 0.889 |
| MRPC | F1 | 0.904 | **0.927** | 0.900 |
| RTE | Accuracy | 0.599 | 0.574 | **0.664** |
| QNLI | Accuracy | 0.908 | **0.910** | 0.905 |
| SST2 | Accuracy | 0.928 | 0.933 | **0.935** |
| MNLI | Accuracy | 0.867 | **0.870** | 0.833 |

Table 5: Evaluation on GLUE Dataset

except one, RTE, where BERT base clearly outperforms the other two. This implies that document corpus pre-training does not necessarily decarease the generalizability of the models. More importantly, we see that structure-aware pre-traning leads to positive effects in some tasks like MRPC, QNLI and MNLI where STRUCTFORMER outperforms both vanilla pre-trained longformer and BERT.

# 6 Discussion and Conclusion

The results in the SciREX evaluation demonstrate that structure awareness positively boosts the performance of the model salient mention clustering and all tasks leading up to it. Moreover, this gain cannot be attributed to contextual learning for two reasons

- SciREX also uses contextual learning before salient mention clustering to train and finetune the coreference model

- The Vanilla Longformer model is also trained on the same dataset as our structure-aware model, and its performance is similar to SciREX baseline.

This, coupled with the fact that attention helps encode keyword information, proves that structure awareness in pre-training helps the model understand the context better for fine-tuning on downstream tasks. The analysis of attention in StructFormer and its differences from vanilla Longformer helps us understand the reasons for its better performance. Moreover, the fact that structure-awareness in pre-training helps the model understand the structure in unseen documents presented as a corpus means that StructFormer can be a better choice for most document-related tasks, without losing out on generalizability, as indicated by the studies on GLUE dataset. Our work clearly highlights that structure-aware pre-training has a positive impact on downstream tasks. We use the global tokens in sparse attention models for pre-training which has not been explored before and demonstrate its advantages over vanilla pre-training for information extraction tasks in long passage documents.

# 7 Limitations

Our work provides the first evidence of using global tokens as a substitute for document structure understanding in pre-training. However, we acknowledge that our findings are preliminary and there is much more to explore in this arena. We have planned a more detailed investigation into attention patterns for future work. This future research would involve a thorough analysis of the model's behavior and attention allocation mechanisms under different contexts, providing a richer understanding of the impacts of structure-aware pre-training. Further, we have pre-trained on only scientific documents, and not on other structured data sources like books. This could significantly help the pre-training process as the model gets wider structural information. Lastly, we have not extended our method to other structured data sources.

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

# A    Appendix

## A.1    Additional figures

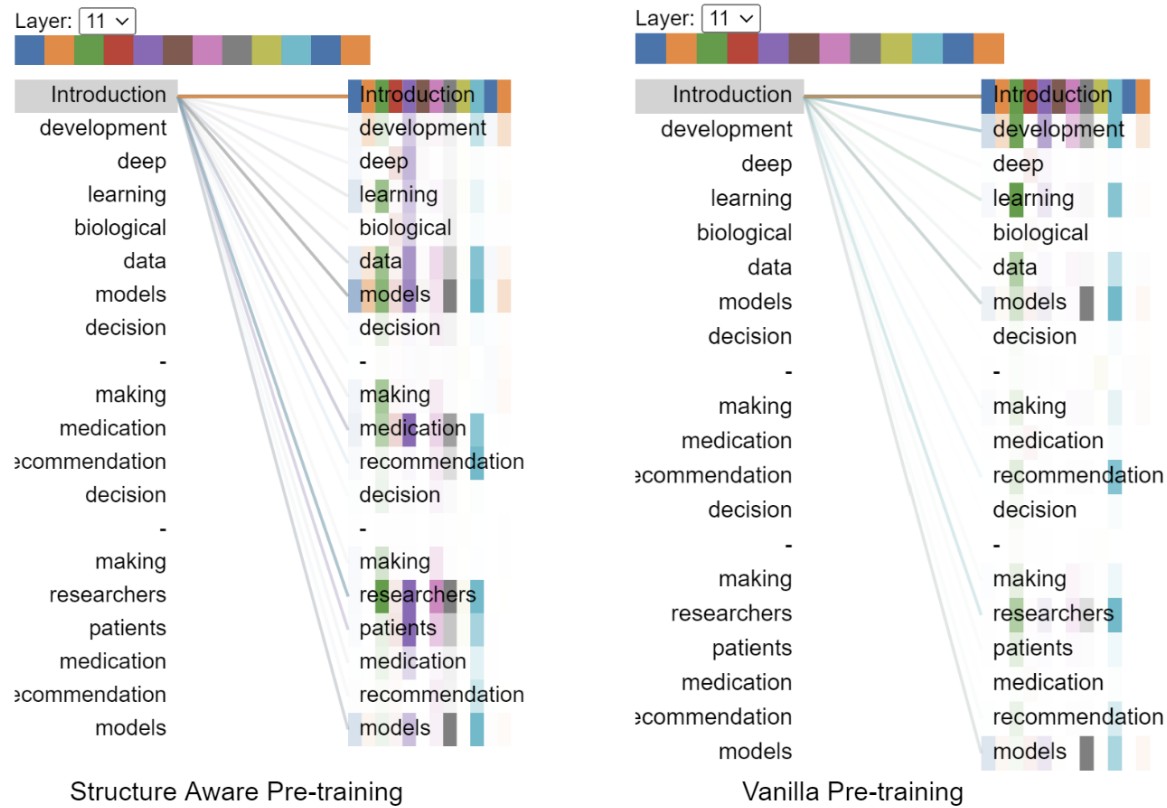

Figure 3: Attention patterns of structure-aware pre-training and vanilla pre-training between header and keywords

## A.2    Document Structure Extraction Example

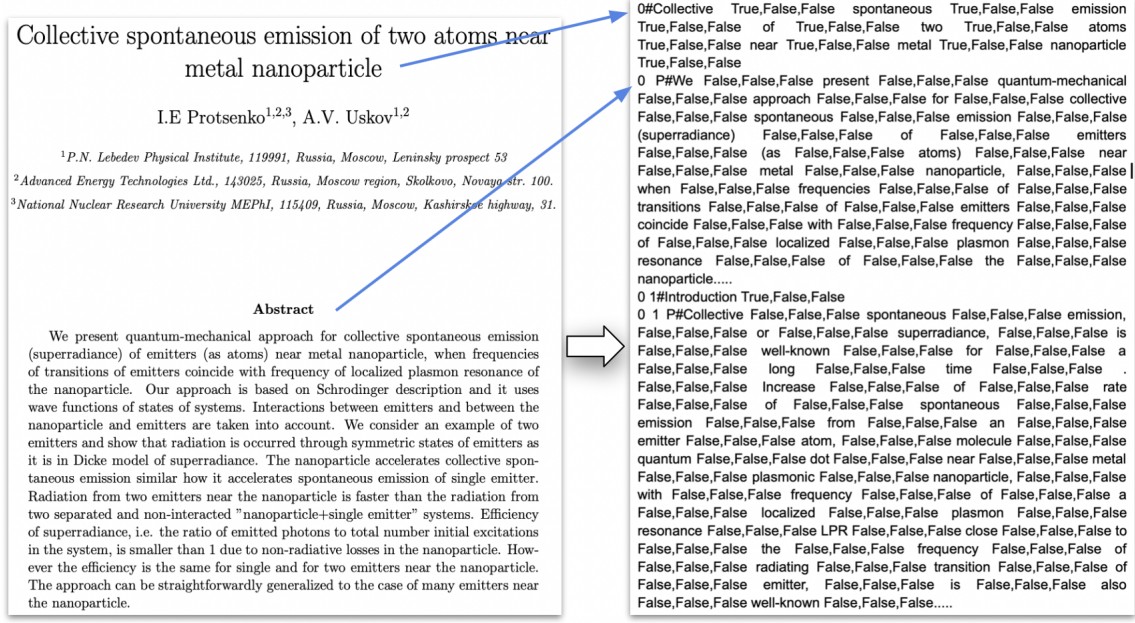

Paper : Igor Protsenko, Alexander Uskov "Collective spontaneous emission of two atoms near metal nanoparticle" (2014).

Figure 4: Example of the extraction and storage of document structure in a text file

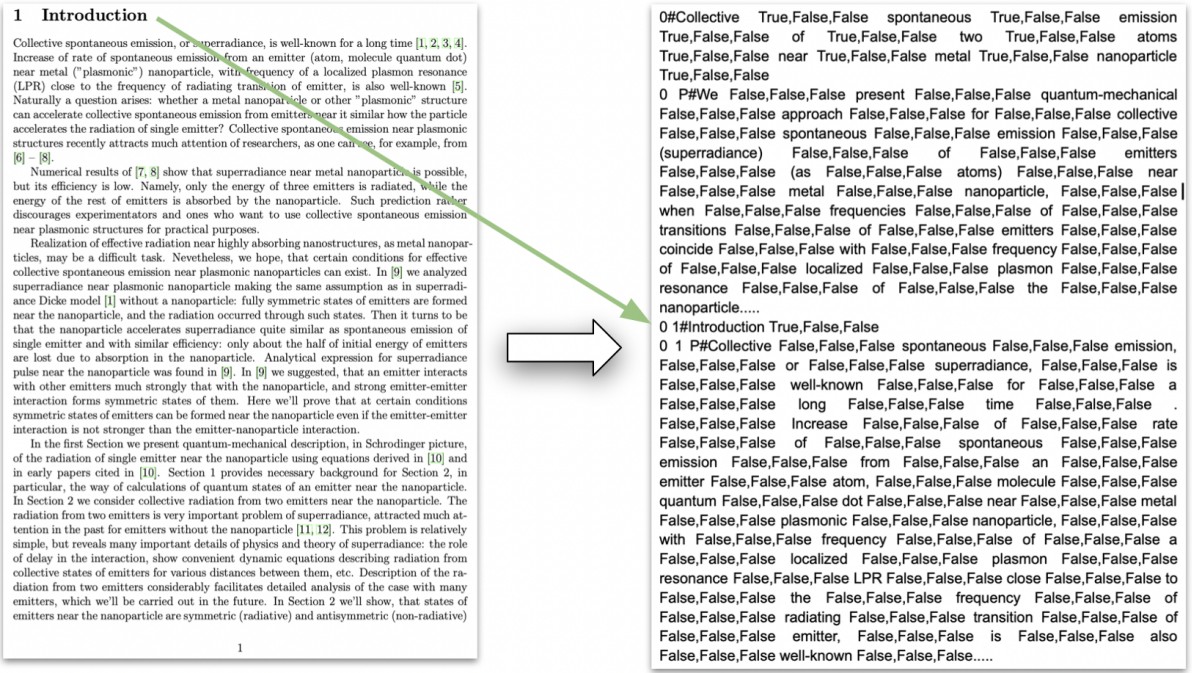

Paper : Igor Protsenko, Alexander Uskov "Collective spontaneous emission of two atoms near metal nanoparticle" (2014).

Figure 5: Example (cont.) of the extraction and storage of document structure in a text file

