# OpenReview forum: "StructFormer: Document Structure-based Masked Attention and its impact on Language Model Pre-Training"
_TMLR — Rejected by TMLR_

### Review · Reviewer_Hop8 · 2025-08-03

**Summary Of Contributions:**

The authors pre-train a LongFormer model on a large dataset of content from arXiv, introducing special global tokens for the headers which attend to both masked and the neighboring tokens in the context window. These interventions are proposed with the stated intention of learning the semantic information of document structure. The authors claim that this approach helps significantly in downstream tasks and provides evidence that learning during pre-training can go beyond natural language understanding.

**Audience:**

No

**Audience Explanation:**

Were they supported by clear evidence, the findings in this paper might be of interest to some individuals in TMLR's audience. As they are not, I cannot see the findings currently contained in the paper being of interest.

**Broader Impact Concerns:**

I see no such concerns.

**Claims And Evidence:**

No

**Claims Explanation:**

* The paper contains numerous typos, errors and misleading statements. For brevity's sake, I will limit myself to page 1.

"While sparse attention mechanisms, akin to full attention in being Turing-complete, have been theoretically established, their practical impact on pre-training remains unexplored." A brief list of large-scale empirical works which have utilized sparse attention in pretraining includes (https://mistral.ai/news/announcing-mistral-7b, https://skillupexchange.com/qwen2-5-a-comprehensive-guide/, https://www.databricks.com/blog/mixattention).

"Typically, the word with the highest probability is selected as the next word ..." Greedy decoding is rarely used in practice; rather, some form of sampling is generally employed (https://heidloff.net/article/greedy-beam-sampling/).

"In the case of attention-based Transformer models, each word learns a self-attention score concerning every other word in the vocabulary V, effectively capturing the relationships between words in a given corpus.". Each token (word/subword) computes attention scores with respect to other tokens in the same input sequence, not with every word in the vocabulary V.

* Despite the extremely long related works, post-2022 work is almost completely absent. Baselines, benchmarks, and points of methodological comparison are almost entirely from 2021 or earlier. The lack of attention to any recent work, especially the extensive literature on generative language modeling, and the absence of strong baselines undercuts the authors' strong claims to significance.

* Experimental results are presented without statistical significance testing; since the benchmarking difference between StructFormer and the baselines is frequently less than 1%, this omission is even more unfortunate than usual.

* Presentation of results are misleading. The paper neglects to boldface that StructFormer is outperformed by SciREX Baseline on Binary Relations. In Table 5, MRPC (the 1 task out of 7 in GLUE where StructFormer significantly outperforms the baselines) gets two metrics where all other tasks get 1.

**Requested Changes:**

Although I think it would be advisable for the authors to consider the above critiques, unfortunately I think the paper is pretty far from a state where I would recommend acceptance. Therefore, I will not request any particular changes.

---

### Review · Reviewer_Q7pX · 2025-08-09

**Summary Of Contributions:**

This paper proposes the StructFormer method, which explicitly incorporates document structure information into the pre-training phase of BERT-like models. It also sets structural tokens such as titles and chapters as global tokens within the Sparse Attention framework (Longformer) to capture the hierarchical structure and semantic organization of documents in Masked Language Modeling tasks. The authors constructed a large-scale structure-aware corpus based on arXiv LaTeX files and compared pre-trained them with text-only versions. They analyzed the role of global tokens in the attention model and their impact on downstream tasks. Experimental results show that this method outperforms the vanilla Longformer on tasks such as salient entity clustering on the SciREX dataset, while maintaining good generalization performance on GLUE tasks.

Strengths:

1. This paper addresses the research gap in sparse attention models that do not explicitly utilize global tokens during pre-training, raising a targeted research question that is worthy of exploration.
2. A large-scale, structure-aware pre-training corpus with structural annotations is constructed.
3. Appropriate controlled experiments are designed, including a comparison of the vanilla Longformer and the structure-aware Longformer, as well as evaluations on SciREX and GLUE tasks, covering both domain-specific and general-purpose tasks.

**Audience:**

Yes

**Audience Explanation:**

This paper explores the effectiveness of explicitly introducing global tokens during the pre-training phase of the Sparse Attention model to leverage document structure information, and conducts systematic comparative experiments on the SciREX and GLUE tasks. TMLR readers, many of whom are researchers working on long text modeling, Transformer architecture improvements, pre-training methods, and document understanding, will be interested in exploring this integration of structured corpus and pre-training strategies.

**Broader Impact Concerns:**

This method is essentially an improvement on the general pre-training strategy. It does not involve the generation of content in sensitive areas and does not directly introduce new ethical risks.

**Claims And Evidence:**

No

**Claims Explanation:**

1. The core conclusion that "structure-aware pre-training significantly improves downstream model performance" is insufficiently supported by evidence. The improvements on the SciREX task are primarily focused on salient clustering, with little improvement seen on other tasks, such as n-ary relation tasks. No statistical significance tests or confidence intervals are provided.
2. The ablation experiments are insufficiently granular. While a comparison is made with the vanilla Longformer, there is a lack of grouping experiments with different types of global tokens (such as title only, title + subsections, etc.), making it impossible to verify which type of structural information contributes most.
3. The attention pattern analysis only provides single examples and average score improvements. It does not quantify the trends across different layers or time steps, nor does it verify whether these pattern changes are directly causal with performance improvements.
4. While the dataset construction and pre-training details (such as hyperparameters and sampling strategies) are described, some key settings (such as the global mask ratio and header detection error rate) are not quantitatively reported, which may affect the reproducibility and credibility of the conclusions.

**Requested Changes:**

1. Add controlled experiments with different global token types and ratios to quantify the independent contributions of different components of structural information.
2. Perform significance tests on SciREX results and report standard deviations or confidence intervals to enhance the statistical robustness of the conclusions.
3. Expand the attention pattern analysis to include statistical results at different Transformer layers and time steps, and attempt to establish a correlation between pattern changes and performance improvements.
4. Report the accuracy and error distribution of the global token detection process and analyze the potential impact of these errors on model performance.

---

### Review · Reviewer_xHpV · 2025-08-18

**Summary Of Contributions:**

This paper, based on arXiv training data, explores the use of global tokens as a substitute for document structure understanding in pre-training, highlighting the importance of incorporating document structure into language models.

**Audience:**

Yes

**Audience Explanation:**

This paper discusses attention mechanisms and structured documents, both of which are important topics in NLP.

**Claims And Evidence:**

No

**Claims Explanation:**

1. Lack of references. Several important concepts and techniques are introduced without proper citation. For instance, in the introduction, mentions of LSTM, BiLSTM, and recent state-of-the-art (SOTA) methods are not supported by references. Similar issues recur throughout the paper. The authors should carefully revise the manuscript to ensure that all key points, algorithms, and claims are properly grounded in prior work.

2. Limited scope of experiments. The evaluation is restricted to arXiv data, which does not adequately support the broad claim that “incorporating document structure into LM models demonstrates their capacity to excel in more abstract tasks, such as document understanding.” To substantiate such claims, the authors should either (a) extend the experiments to more diverse structured data sources (e.g., code), or (b) moderate the strength of their claims accordingly.

3. Unclear motivation and limited novelty. The motivation for studying the influence of global attention on BERT pre-training is not sufficiently clear, apart from the remark that “the impact of global tokens during the pre-training phase has not been thoroughly explored.” The paper does not present a new problem formulation, method, or particularly strong insights. Furthermore, it does not explain why StructFormer outperforms the Vanilla approach in structural learning and generalization. Clarifying these aspects would strengthen the contribution.

**Requested Changes:**

Please refer to above comments.

---

### Decision · Action_Editor_pALe · 2025-10-04

**Recommendation:** Reject

**Additional Comments:**

Several weaknesses and questions came up in the reviews. The authors never responded during the discussion period, and therefore from the issues brought up by the initial reviews remain. Therefore, the recommendation is to reject the current submission. However, the topic itself is of interest and the angle of approaching it is creative and could be valuable. I encourage the authors to review the comments and revise the paper accordingly for future submissions.

**Audience:**

Yes

**Audience Explanation:**

The topic of integrating document structure into language models, particularly for long-text understanding, is highly relevant and of interest to the TMLR audience. However, the paper needs to go through major revision before others can benefit from it.

**Claims And Evidence:**

No

**Claims Explanation:**

The consensus among the reviewers is that the paper's claims are not adequately supported. The primary issues are:

* Insufficient Empirical Support: The central claim that "structure-aware pre-training significantly improves downstream model performance" appears to be overstated. As noted by Reviewer Q7pX, the performance gains are concentrated on a single sub-task (salient clustering on SciREX) and lack statistical significance tests or confidence intervals to validate their robustness, a concern also raised by Reviewer Hop8.
* Limited Scope of Experiments: The evaluation is restricted to arXiv data, which, as Reviewer xHpV points out, is not sufficient to support the broad claims about general "document understanding."
* The paper contains several inaccurate statements around the fundamentals of Transformers and attention, and fails to engage with relevant work post-2022 and lacks proper citations for several key concepts.

**Resubmission Of Major Revision:**

The authors may consider submitting a major revision at a later time.